# Inclusion of participants from low-income and middle-income countries in core outcome sets development: a systematic review

Jamlick Karumbi  ,[1,2] Sarah L Gorst  ,[1] David Gathara,[2,3] Elizabeth Gargon,[1] Bridget Young,[1] Paula R Williamson[1]

[1]Health Data Science, University of Liverpool, Liverpool, UK
[2]Health Systems Research, KEMRI-Wellcome Trust Research Programme Nairobi, Nairobi, Kenya
[3]Centre for Maternal, Adolescent, Reproductive & Child Health (MARCH), London School of Hygiene & Tropical Medicine, London, UK

**Correspondence to**
Dr Jamlick Karumbi;
J.Karumbi@liverpool.ac.uk

## ABSTRACT

**Objective** Our study aims to describe differences or similarities in the scope, participant characteristics and methods used in core outcome sets (COS) development when only participants from high-income countries (HICs) were involved compared with when participants from low-income and middle-income countries (LMICs) were also involved.

**Design** Systematic review.

**Data sources** Annual Core Outcome Measures in Effectiveness Trials systematic reviews of COS which are updated based on SCOPUS and MEDLINE, searches. The latest systematic review included studies published up to the end of 2019.

**Eligibility criteria for selecting studies** We included studies reporting development of a COS for use in research regardless of age, health condition or setting. Studies reporting the development of a COS for patient-reported outcomes or adverse events or complications were also included.

**Data extraction and synthesis** Data were extracted in relation to scope of the COS study, participant categories and the methods used in outcome selection.

**Results** Studies describing 370 COS were identified in the database. Of these, 75 (20%) included participants from LMICs. Only four COS were initiated from an LMIC setting. More than half of COS with LMIC participants were developed in the last 5 years. Cancer and rheumatology were the dominant disease domains. Overall, over 259 (70%) of COS explicitly reported including clinical experts; this was higher where LMIC participants were also included 340 (92%). Most LMIC participants were from China, Brazil and South Africa. Mixed methods for consensus building were used across the two settings.

**Conclusion** Progress has been made in including LMIC participants in the development of COS, however, there is a need to explore how to enable initiation of COS development from a range of LMIC settings, how to ensure prioritisation of COS that better reflects the burden of disease in these contexts and how to improve public participation from LMICs.

## BACKGROUND

A key barrier to translating clinical research into practice is lack of conclusive evidence.

### Strengths and limitations of this study

► This is the first paper describing the differences and similarities in the core outcome sets (COS) development processes when there are participants from low-income and middle-income countries (LMICs) compared with when participants are exclusively from high-income country settings.

► This paper describes COS published up to 2019. It is possible that more COS involving LMICs participants have been developed during 2020 or are in the process of being developed now.

► Public participation has been described in only a few studies, this may introduce a possible bias in the classification of public participants categories thereby altering the conclusions of public participation.

This is due to different trials, of the similar interventions, being too heterogenous and reporting on outcomes that are often non comparable[1 2] or not relevant to patients and users.[3 4]

Development of core outcome sets (COSs) could ensure all future research in a field reports a common subset of outcomes. This would reduce research waste thereby enhancing comparability and improving research translation and use.[5 6] COSs are agreed-on minimum standardised outcome sets that should be measured and reported in all research in a given health area.[7] They consist of a core domain set (this defines what domains should be measured) and core outcome measurement set (defines the instruments which would be appropriate to measure the domain).[8] In recent years, the use of COS has been promoted by journals, Cochrane review groups and funders.[9 10] COSs have also become useful in routine clinical practice data collection for clinical audit and feedback.[11 12]

Since 2010, the Core Outcome Measures in Effectiveness Trials (COMET) Initiative has been collating, stimulating and promoting the development and use of COS by maintaining an up to date publicly available searchable database.[13] The COMET database includes published studies of COS development, as well as planned and ongoing work. The types of studies included in the database are those in which COS have been developed, as well as studies relevant to COS development, including systematic reviews of outcomes. Within the database, there are publications that describe a COS only (hereafter denoted specific for COS), and other publications that describe a COS as well as provides recommendations about other trial design aspects like eligibility criteria (hereafter denoted as COS as part of a wider trial design). Studies describing the development of a patient-reported outcome (PRO) COS (a core set of patients reported symptoms and health-related quality of life domains) or core event set (a core set of adverse events or complications). Initially, the database included studies that had been identified by ad hoc means, however, it is now systematically updated using the annually updated systematic reviews as the basis of the updates since 2013.[14–17] The use of this database should help minimise duplication of effort in the development of COS and potentially encourage the use of COS to tackle global health challenges.

## HOW ARE COS DEVELOPED?

There is no gold standard for development of COS, however minimum standards have been agreed in the last few years. These are the Core Outcome Set-STAndards for Development (COS-STAD) recommendations, comprising 11 minimum standards for all COS development projects.[18] The recommendations focus on three key domains: (1) the scope of the COS; specific area of health or healthcare that the COS applies needs to be described, with details of health condition, population and types of interventions. The COS may be developed to encompass all stages or severity of a health condition or it may be focused on a particular sub-population, for example, a COS may be developed for all patients with COVID-19, or it may focus on patients with long COVID-19 only. (2) the stakeholders involved in the COS development; the stakeholder groups to be involved and the target number from each group is dependent on the scope of the COS and the existing knowledge and practicability. Decisions on stakeholders' involvement should be documented and explained in the study protocol. (3) the consensus processes; this typically involves developing a consensus on 'what', 'how' and 'when' to measure. Some of the methods used to build consensus include: Delphi technique; nominal group technique, consensus development conference and semistructured group discussion. The Delphi technique with two or three rounds combined with a consensus conference has been most widely used.[19–21] Choice of method is informed by several factors including; the need to build a true consensus with methodological rigour, strategies to ensure that a diverse range of opinions are heard, and

factors such as financial and carbon costs that might limit the practicality of face-to-face meetings.[7]

To date, COSs have been developed for various thematic or disease conditions and more are being developed.[13 19] It has been opined that development and use of a COS would help reduce outcome heterogeneity and reporting bias, while ensuring that wide-ranging perspectives, including patients' opinions, are incorporated, thus enhancing the value and quality of research.[22] Most COSs have been developed from the perspectives of high-income countries (HICs), with over 70% of COS including participants from Europe and North America. There has been an increase in the proportion of COS that have included participants from Africa and South American countries in COS development, but this still remains low at around 25% as per COS published up to the end of 2018.[19] A closer look shows that the proportion of COS participants residing in low-income and middle-income countries (LMICs) of these regions is even lower being just about 16% in 2016. Further, participants have mainly been those with research and clinical expertise in the various disease areas rather than as public participants with relevant lived experience.[16]

Previous work describing COS studies[14–16 19] has not examined whether there is a difference in the scope, stakeholder participation and approaches to consensus between COS that have include had participants only from HICs compared with those that also included participants from LMICs. The differences in the global burden of diseases may influence which COS are developed, and available resources may determine the methods used in COS development. For example, LMICs have had a higher level of infectious diseases while non communicable diseases have been generally more prevalent in HICs. The hypothesis therefore is that COS with LMIC stakeholders are more likely to be developed for infectious disease conditions and those with predominately HIC stakeholders to are likely to be for non-communicable diseases. Additionally, different methodologies for consensus building require different level of resources, for example, a face-to-face Delphi workshop may be more expensive if it must include a wider range of stakeholders and online Delphi processes are also likely to be influenced by internet connectivity which may be lower in LMICs compared with HICs. The aim of this paper is to describe the differences or similarities in the scope, participant characteristics and methods used in COS developed with participants from HICs and LMICs as classified by the Organisation for Economic Co-operation and Development.[23]

## METHODS
### Study selection
#### Inclusion criteria

We included all COS studies, identified from the original COMET systematic review and annual updates.[14–17 24 25] Studies were eligible for inclusion in the review if they developed or applied methodology for determining which outcome domains or outcomes should be measured in research clinical trials or other forms of health. Studies

were eligible for inclusion if they reported the development of a COS, regardless of any restrictions by age, health condition or setting. Studies describing the development of a PRO COS (a core set of patients reported symptoms and health-related quality of life domains) or core event set (a core set of adverse events or complications) were also eligible for inclusion. Studies describing the update of an existing COS are included as linked papers to the original COS.[17] Eligible studies are added to the database, as they are found with the annual update to the systematic review, ensuring the database is kept up to date.

## Identification of relevant studies

The search strategy for identifying eligible COS development studies has previously been described[14–17] and is briefly described here. A comprehensive search is used to identify studies that had been published or indexed in SCOPUS and MEDLINE via Ovid from inception up to December 2019. The search strategy, which was developed for the original COMET systematic review in 2013, is provided in online supplemental appendix. Annual database searches were repeated in 2015–2020 to identify

studies published up to the end of 2019. Hand searching of studies is also performed.

The review was conducted in accordance with the Preferred Reporting Items of Systematic Reviews and Meta-Analyses (PRISMA) guidelines and the included studies are shown in a PRISMA flow diagram (see figure 1).

## Data extraction

Data were extracted in relation to scope of the COS study (aim of the study, intended use of the COS, year of publication, target population, intervention and disease or condition area), participant categories (clinicians, researchers etc, and their geographic location) and the methods used in outcome selection (Delphi, interviews, focus group discussions, etc, alone or in combination). Data on the burden of disease in LMICs for the last 5 years were obtained, from the Institute of Health Metrics and Evaluation database,[26] to help check its alignment to the COS developed over the same period.

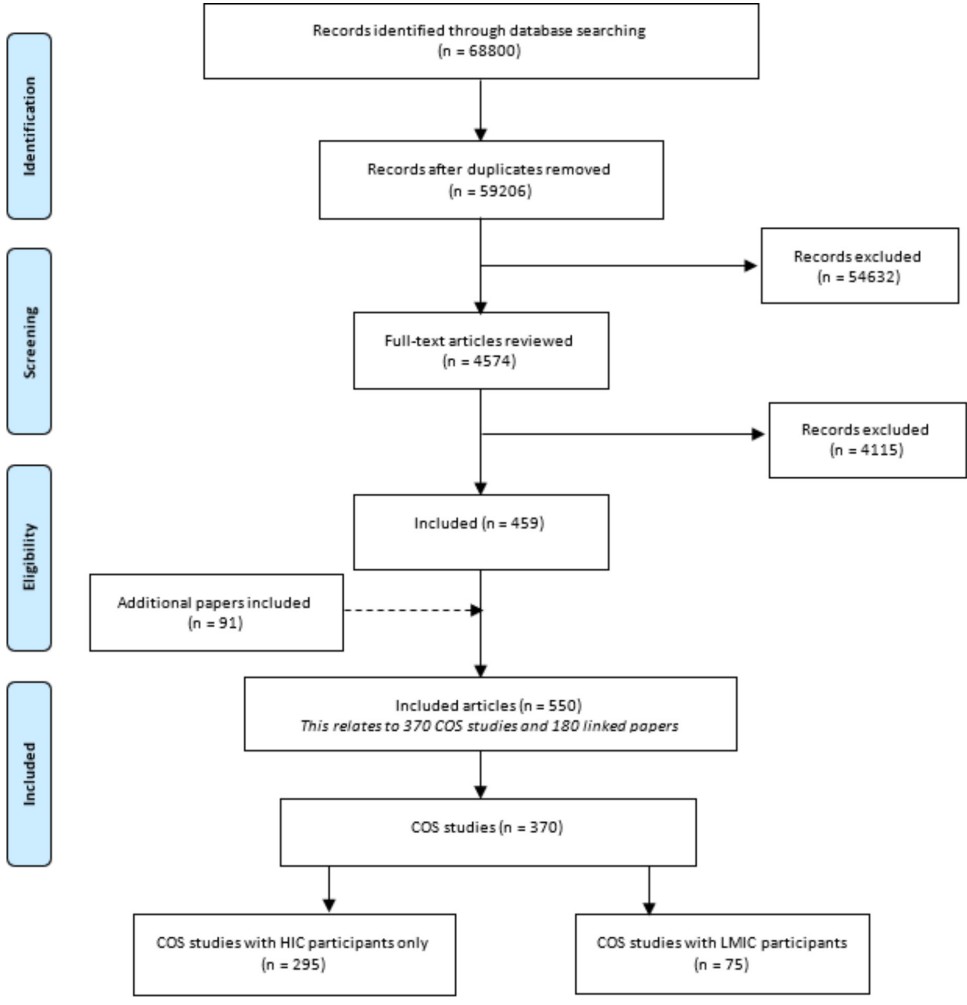

**Figure 1** Preferred Reporting Items for Systematic Reviews and Meta-Analyses flow chart of identification of eligible studies from the COMET database. Data were extracted from the COS systematic reviews.[14–17] COS, core outcome sets; COMET, Core Outcome Measures in Effectiveness Trial; HIC, high-income country; LMICs, low-income and middle-income countries.

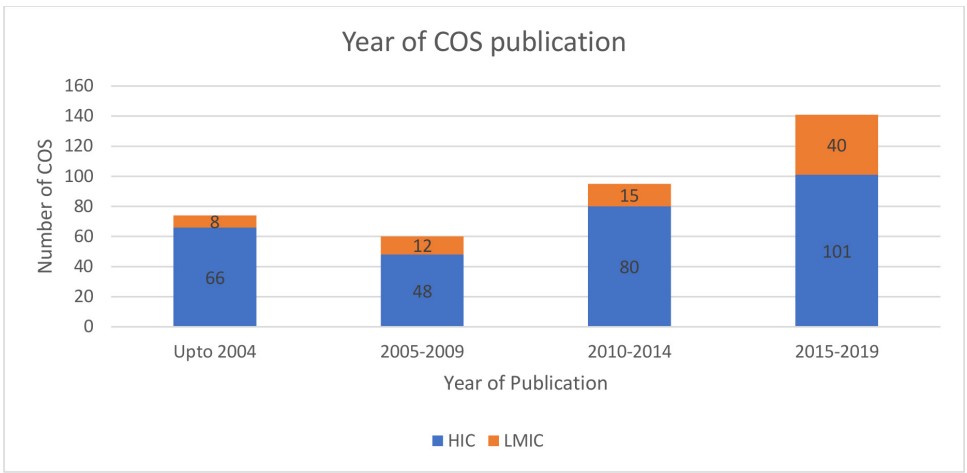

**Figure 2** Year of COS publication. COS, core outcome sets; HIC, high-income country; LMIC, low-income and middle-income country.

## Analysis

A description of the proportion of COS that included participants from LMICs and those which were initiated from the LMIC setting is provided. For a COS to be classified as being initiated from an LMIC, either ethical approval of the study had to have been sought in an LMIC, and/or the affiliation of the lead author had to be based in an LMIC; we also used the sites of participant recruitment as confirmation. The scope, participant categories and consensus methods used are described for COS that included participants from LMICs and those with exclusively HIC participants separately. Participation here is defined as being a study participant and contributing data to the COS study.

## PATIENTS AND PUBLIC INVOLVEMENT

Patients and the public were not involved in the development of the research question or the design and conduct of this systematic review.

## FINDINGS

### General description

There was a total of 550 references published by end of 2019 in the COMET Initiative database describing 370 COS. Of these 370 COS, 295 (80%) included only HIC participants and 75 (20%) also included LMIC participants. Two COS had only LMIC participants, one in Sri Lanka[27] and another in China.[28] A flow diagram of the study selection process is provided in figure 1 as guided by PRISMA statement.[29]

### Year of publication

Between 2014 and 2019, there has been an almost twofold increase in COS that have included LMIC participants to 28% (40/141) from 15% (35/229) in preceding years. Figure 2 describes the absolute numbers of the COS developed up to 2019.

## Scope of included studies

The comparison of the scope of the included studies is shown in table 1. Overall, more than half (63%) of the studies were specifically designed for COS development rather than being part of a wider project or trial. However, a higher proportion of studies with LMIC participants were specific for COS (83%) compared with those with only HIC participants (58%). Over 80% of COS were developed for research only. More than half of studies (55%) did not explicitly report on the age group of the target population characteristics. During the interpretation of the data, this was assumed to be adults. This was based on the disease being predominantly associated with the adult population and the lack of reference to any literature on children.

Cancer (16% for COS with HIC participants only and 12% for COS with LMIC participants) and rheumatology (9% for COS with HIC participants only and 16% for COS with LMIC participants) were the dominant disease domains across the COS developed either with LMIC or HIC participants. A slightly higher proportion of COS for rheumatology, pregnancy, skin, anaesthesia and child health included participants from LMICs compared with HICs. Five COS on health systems had LMIC participants. Figure 3 describes the disease categories that were covered by the COS.

Of the 75 COS with LMIC participants, only four (5%) were initiated from LMIC setting. All were from Asia, with two from China[28 30] and one each from Sri Lanka[27] and the Philippines.[31] No COS has been initiated from Africa or South America, although COS for conditions which are more prevalent in LMIC settings had a higher number of participants from LMICs. For example, the malaria COS by Moorthy *et al*[32] included participants from ten sub-Sahara African Countries but none from other LMICs. The COS for the prevention of preterm births had LMIC participants from Egypt, Nigeria, South Africa, China, Iran, Lebanon, Pakistan, Argentina and Brazil.[33]

| Table 1 | Scope of included studies | | |
|---|---|---|---|
| | HICs n (%) (N=295) | LMICs n (%) (N=75) | Total (%) (N=370) |
| **Scope of the COS study** | | | |
| Study aims | | | |
| Part of wider trial design* | 124 (42) | 13 (17) | 137 (37) |
| Specific for COS† | 171 (58) | 62 (83) | 233 (63) |
| Intended use of recommendations | | | |
| Research | 264 (89) | 61 (81) | 325 (88) |
| Research and practice | 31 (11) | 14 (19) | 45 (12) |
| Population characteristics | | | |
| Neonates | 4 (1) | 1 (1) | 5 (1) |
| Adults | 72 (24) | 16 (21) | 88 (24) |
| Children | 27 (9) | 7 (9) | 34 (9) |
| Children and adults | 28 (9) | 10 (13) | 38 (10) |
| Not specified‡ | 164 (56) | 41 (55) | 205 (55) |
| Intervention characteristic | | | |
| Any intervention | 77 (26) | 30 (40) | 107 (29) |
| Device | 5 (2) | 1 (1) | 6 (2) |
| Device and surgery | 7 (2) | 1 (1) | 8 (2) |
| Pharmacological treatment | 50 (17) | 9 (12) | 59 (16) |
| Procedure | 9 (3) | 5 (7) | 14 (4) |
| Radiotherapy | 2 (1) | 0 (0) | 2 (1) |
| Rehabilitation | 9 (3) | 2 (3) | 11 (3) |
| Screening | 2 (1) | 0 (0) | 2 (1) |
| Surgical | 31 (11) | 6 (8) | 37 (10) |
| Vaccine | 0 (0) | 2 (3) | 2 (3) |
| Both pharmacological and pharmacological treatment | 3 (1) | 0 (0) | 3 (1) |
| Non-pharmacological treatment | 4 (1) | 2 (3) | 6 (2) |
| Health system | 5 (2) | 2 (3) | 7 (2) |
| Not specified | 91 (31) | 15 (20) | 106 (29) |

*Publications that describe a COS as well as provides recommendations about other trial design aspects like eligibility criteria.
†Publications that describe a COS only.
‡Interpretated to mean adults.
COS, core outcome set; HICs, high-income countries; LMICs, low-income and middle-income countries.

### Characteristics of participants involved in consensus process

All COS that had LMIC participants explicitly reported characteristics of their sample compared with 92% of COS that had only HIC participants (see table 2). Overall, over 70% of COS explicitly reported including clinical experts; this was higher where LMIC participants were involved (92%). A slightly higher proportion of COS that included LMIC participants (57%) had included members of the public compared with those with HIC participants only (32%). A higher proportion of LMIC COS (23%) had carers as part of public participants compared with the HIC COS (9%). Of the 28 COS with LMIC participants, only 4 (all with HIC patient participants too) translated the COS development materials to non-English languages. These are COS for clinical trials in acute diarrhoea,[34] infant colic,[35] childhood constipation[36] and type 2 diabetes.[37] Non-clinical research expertise was more common in COS with LMIC participants (44%) compared with those with HIC participants only (31%).

Participants from LMICs were drawn from the three continents of Asia (60%), South America (49%) and Africa (31%). However, there was a skewed representation with LMIC participants largely being from three countries, with South Africa representing 50% of participants from Africa, Brazil representing 46% of participants from South America and China representing 30% of the participants from Asia.

### Methods for selection of outcomes

Methods used for outcome selection ranged from Delphi, consensus conferences and semistructured discussion as shown in table 3. Rarely was one method used for COS development. Semistructured discussion seemed to be the most commonly used single method, with 20% of COS with HIC participants only and 8% of COS with LMIC participants describing its usage.

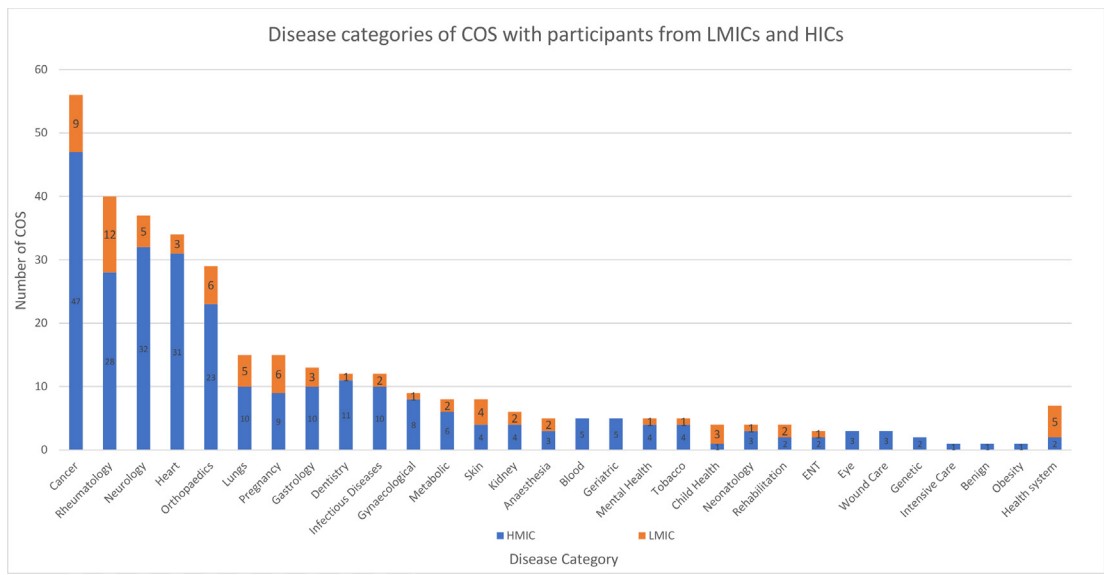

**Figure 3** Comparison of disease categories covered in the COS with exclusively HIC participants and those also with LMIC participants. COS, core outcome sets; HIC, high-income country; LMIC, low-income and middle-income country.

Most COS used a combination of methods. Literature review in combination with other methods was most common, with 37% of COS with HIC participants and 60% of COS with LMIC participants using this method as shown in table 3. Only one COS that included LMIC participants had no methods described compared with six that only had HIC participants.

A combination of the three most common methods was more likely to be used for LMIC COS (n=23, 30%) compared with the HIC COS (n=37, 13%) as shown in figure 4.

## DISCUSSION

This review provides the first comparison of COS that have been developed with participants from HICs and LMICs.

Out of the 370 published COS, 75 (20%) COS included participants from LMICs. Thirty-three COS have been published since the fifth update of COS systematic review with fifteen including participants from LMICs.[19] The COS span a range of health areas, target populations and methods used. The methods for outcome selection are similar across the settings, perhaps reflecting the increased standardisation of methods of COS development generally.

Interestingly, even though participants from LMICs have been involved in earlier COS, for example, the cancer reporting recommendations by Miller and others in 1981,[38] the inclusion of LMIC participants has increased predominantly in the last 5 years. In this time, there has been an almost twofold increase in the proportion of COS that have had LMIC participants. Global connectivity is allowing people to conduct research without the need for physical meetings. A look at the development processes used is 'older' COS shows that most depended on consensus building during conferences and physical

workshops.[32 38 39] These kinds of consensus meetings tend to be more expensive and limit inclusion of participants from varied settings. Despite the increase, Brazil, China and South Africa have been the main drivers of participation from LMICs.

All the COS developed across the settings have mainly focused on research or research and practice. A slightly higher proportion (19%) of COS with LMIC participants are for research and practice compared with 11% of COS with HIC participants only. Given that COS with participants from LMICs are 'newer' this could reflect that more recent COS are being developed with the intention of spanning the continuum from research to routine practice in mind and also recent availability of guidance for COS development like the COMET handbook[11] and COS-STAD[18] might have influenced the inclusion of wider groups of stakeholders.

Although the COS span a wide range of conditions, the top five conditions where COS exist are cancer, rheumatology, neurology, cardiovascular and orthopaedic conditions. These are mainly non communicable diseases, and even though these are now becoming more common in LMICs, the burden of these conditions has mainly been in HICs.[26] Of the COS with LMIC participants, pregnancy, lungs and health systems were among the top five conditions. LMICs are still facing infectious diseases like neonatal sepsis, lower respiratory tract infections, malaria and HIV among others,[26 40] and one would therefore expect that COS with LMIC participation would reflect this. Only two COS, for malaria[32] and leprosy[31] represent conditions that are usually of high burden in LMICs. Thus, even though LMIC participants are being included in COS development, COS for research may not be reflecting the disease burden in LMICs. This may be because research funding priorities, for which COS are then being developed, are more likely to reflect an HIC perspective.

**Table 2** Participant characteristics

| Participant category | Subcategory (not mutually exclusive) | HIC n (%*) | LMIC n (%*) | Total (%) |
|---|---|---|---|---|
| Clinical experts | | 201/295 (68) | 69/75 (92) | 270/370 (73) |
| | Clinical experts | 166 (56) | 62 (83) | 228 (62) |
| | Clinical research experts | 92 (31) | 41 (55) | 133 (36) |
| | Members of a clinical trial network | 8 (3) | 7 (9) | 15 (4) |
| | Clinical and non-clinical researchers | 12 (4) | 1 (1) | 13 (4) |
| Public participation | | 94/295 (32) | 43/75 (57) | 137/370 (37) |
| | Patients | 72 (24) | 28 (37) | 100 (27) |
| | Carers | 26 (9) | 17 (23) | 43 (12) |
| | Patient support group representatives | 21 (7) | 9 (12) | 30 (8) |
| | Service users | 4 (1) | 5 (7) | 9 (2) |
| Non-clinical research expertise | | 92/295 (31) | 33/75 (44) | 125/370 (34) |
| | Researchers | 55 (19) | 21 (28) | 76 (21) |
| | Statisticians | 22 (7) | 8 (11) | 30 (8) |
| | Epidemiologists | 13 (4) | 9 (12) | 22 (6) |
| | Academic representatives | 5 (2) | 0 (0) | 5 (1) |
| | Methodologists | 19 (6) | 5 (7) | 24 (6) |
| | Economists | 7 (2) | 2 (3) | 9 (2) |
| Authorities | | 50/295 (17) | 22/75 (29) | 72/370 (19) |
| | Regulatory agency representatives | 33 (11) | 15 (20) | 48 (13) |
| | Government agencies | 14 (5) | 5 (7) | 19 (5) |
| | Policy makers | 10 (3) | 7 (9) | 17 (5) |
| | Charities | 4 (1) | 0 (0) | 4 (1) |
| | Service commissioners | 3 (1) | 1 (1) | 4 (1) |
| Industry representatives | | 36/295 (12) | 19/75 (25) | 55/270 (20) |
| | Pharmaceutical industries | 32 (11) | 19 (25) | 51 (14) |
| | Device manufacturers | 4 (1) | 1 (1) | 5 (1) |
| | Biotech company representatives | 1 (1) | 0 (0) | 1 (1) |
| Others | | 10/295 (3) | 8/75 (11) | 18/370 (5) |
| | Service providers | 5 (2) | 2 (3) | 7 (2) |
| | Ethicists | 1 (1) | 1 (1) | 2 (1) |
| | Journal editors | 4 (1) | 5 (7) | 9 (2) |
| | Systematic review authors | 13 (4) | 2 (3) | 15 (4) |
| Not stated | Not reported | 23 (8) | 0 (0) | 23 (6) |

*Since the categories are not mutually exclusive each percentage is based on the total for that category, that is, 295 for HIC, 75 for LMIC and 370 for the total.
HIC, high-income country; LMIC, low-income and middle-income country.

Interestingly, only four COS have been initiated from an LMIC setting, two from China (COS for hip-preserving treatment of osteonecrosis of the femoral head,[28] COS for infertility treatment research[30]) and one each in Sri Lanka (COS for effectiveness of antiepileptic therapy in children[27]) and the Philippines, (COS for tibialis posterior transfer surgery[31]). There is still no COS initiated from Africa or South America. Given the varying disease epidemiology and burden between HICs and LMICs there is a need to identify mechanisms to enable LMIC stakeholders to develop context relevant COS so as to improve applicability and adoptability of COS to these settings.[14] These strategies could probably include awareness raising about COS in LMIC settings, provision of resources for COS development and dissemination of guidelines and methods for COS development. These are some of the barriers which have been described potentially affecting COS development from an HIC perspective (interviewees were mainly from Europe and North America).[41] There is a need to assess which barriers and

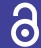

**Table 3** Methods used in the selection of outcomes

| Description | HIC n=295 (%) | LMIC n=75 (%) | Total n=370 (%) |
|---|---|---|---|
| Delphi only | 10 (3) | 5 (7) | 15 (4) |
| Consensus conference only | 11 (4) | 4 (5) | 15 (4) |
| Focus group discussions only | 1 (<1) | 0 (0) | 1 (<1) |
| Nominal group techniques only | 1 (<1) | 0 (0) | 1 (<1) |
| Semistructured discussions only | 64 (22) | 6 (8) | 70 (19) |
| Survey only | 4 (1) | 1 (1) | 5 (1) |
| Review of literature only | 32 (4) | 4 (5) | 36 (10) |
| Interviews only | 1 (<1) | 0 (0) | 1 (<1) |
| **Mixed methods** | 171/295 (58) | 55/75 (73) | 226/370 (61) |
| Consensus conference and other(s) | 8 (3) | 1 (1) | 9 (2) |
| Delphi and other(s) | 73(25) | 42 (56) | 115 (31) |
| Focus group discussions and other(s) | 7 (2) | 7 (9) | 14 (4) |
| Nominal group techniques and other(s) | 15 (5) | 11 (15) | 26 (7) |
| Semi structured discussions and other(s) | 88 (30) | 33 (44) | 121 (33) |
| Survey and other(s) | 25 (8) | 8 (11) | 33 (9) |
| Review of literature and other(s) | 108 (37) | 45 (60) | 153 (41) |
| Interviews and other(s) | 20 (7) | 7 (9) | 27 (7) |
| Unstructured and other(s) | 4 (1) | 8 (11) | 12 (3) |
| No methods described | 6 (2) | 1 (1) | 7 (2) |

HIC, high-income country; LMIC, low-income and middle-income country.

enablers are particularly relevant to COS development in LMIC settings.

Details of public participation are still suboptimally reported, and where reported, a higher proportion of COS with LMIC participants had some public participation compared with COS with HIC participants only. Of note however, is that public participants tended to be from HIC settings. For example, the type 2 diabetes COS included public participants from the UK and Greece,[37] the breast cancer COS included patients with breast cancer from USA[42] and the COS for acute diarrhoea,[34] infant colic,[35] childhood constipation[36] all included public participants from Europe. Public participation from LMICs could potentially be improved by translation of consensus building materials into non-English languages. Harman et al translated Delphi materials into Portuguese language in a two times a day to improve public participation from Brazil.[37] Unfortunately, there still were no public participants from Brazil suggesting that there are additional reasons for the lack of public participants from LMIC settings.

The lack of public participation from LMICs could be due to numerous reasons some of which could include temporal, cultural and resources differences between

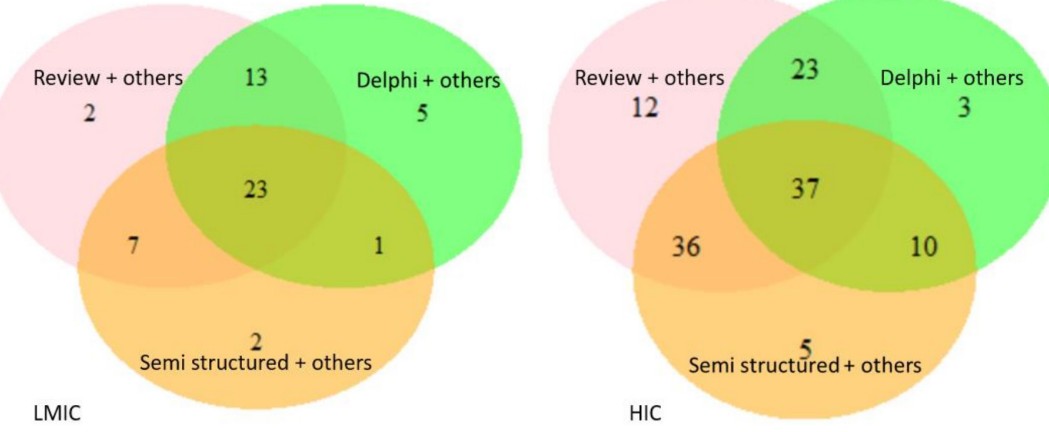

**Figure 4** Combinations of methods for outcome selection. HIC, high-income country; LMIC, low-income and middle-income countries.

LMICs and HICs. First, COS that have included participants from LMICs have mostly been developed in the last decade, during which guidance has been published on how to develop COS including patient participation. Second, the availability of organised patient groups in HIC settings for given conditions or diseases provides a means of identifying and inviting patients as public participants; such groups are generally lacking in LMIC settings. Third, information asymmetry between the public and patients versus clinicians or researchers could potentially limit the participation of the former in COS development. Although evidence is scarce, we suggest that information asymmetry may be a bigger problem in LMICs due to lower literacy levels when compared with HICs.[43] Additionally, some of the major research funders in HICs, for example, the National Institute for Health Research in the UK and the Patient-Centred Outcomes Research Institute in the USA, are increasingly championing public participation in research.

It is therefore likely that a multipronged approach is needed. Further research to explore what would work, in addition to language translation, to improve public participation from LMICs is also needed.

Additionally, there is need to describe the stage at which public participants are involved in COS development and whether it influences which outcomes are included in the COS in a given area. Previous work has shown that patients might sometimes be inhibited when put together with clinicians in a consensus building process[44] but recent experiences show this is changing, with high satisfaction reported, at least from HIC countries, where joint consensus meetings are held.[37 45]

This is the first paper describing the differences and similarities in the COS development processes when there are participants from LMICs compared with when participants are exclusively from HIC settings. However, there are a few limitations. First, this paper describes COS published up to 2019. It is possible that more COS involving LMICs participants have been developed during 2020 or are in the process of being developed now. Second, public participation has been described in only a few studies, this may introduce a possible bias in the classification of public participants categories thereby altering the conclusions of public participation.

## CONCLUSION

Progress has been made in including LMIC participants in the development of COS. Most COS are still being developed with an HIC perspective and as such, still reflect the priorities from HIC perspectives. Only four COS have been initiated from LMICs. Public participation is still low and is poorly documented. There is, therefore, a need to explore how to enable initiation of COS development from a range of LMIC settings, how to ensure prioritisation of COS that better reflects the burden of disease in these contexts and how to improve public participation from LMICs.

**Acknowledgements** This work is also published with the permission of the Director of KEMRI.

**Contributors** Study conception, design and planning of the study were by JK, PRW, DG, BY, SLG, EG as this was part of JK's PhD. Data aggregation, analysis and writing the original draft were by JK. JK screened all title and abstracts supported by PRW and SLG who carried out double screening of these articles. Full-text articles were screened by JK and any uncertainties were discussed with the team. PRW is fully responsible for the overall content for the work and the conduct of the study, had access to the data, and controlled the decision to publish. All authors were involved with reviewing and editing the manuscript.

**Funding** JK has been supported to undertake this work as part of a PhD studentship from the University of Liverpool within the MRC/NIHR Trials Methodology Research Partnership.

**Competing interests** None declared.

**Patient consent for publication** Not applicable.

**Provenance and peer review** Not commissioned; externally peer reviewed.

**Data availability statement** All data relevant to the study are included in the article or uploaded as online supplemental information.

**ORCID iDs**
Jamlick Karumbi http://orcid.org/0000-0003-0848-7821
Sarah L Gorst http://orcid.org/0000-0002-7818-9646

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
