## [Reviewer comments · BMJ Open]

ARTICLE DETAILS

TITLE (PROVISIONAL)	Inclusion of participants from low-and middle-income countries in Core Outcome Sets development: a systematic review
AUTHORS	Karumbi, Jamlick; Gorst, Sarah L.; Gathara, David; Gargon, Elizabeth; Young, Bridget; Williamson, Paula

VERSION 1 – REVIEW

REVIEWER	Brown, Victoria Deakin University Faculty of Health, Deakin Health Economics
REVIEW RETURNED	31-Mar-2021

GENERAL COMMENTS	The paper provides a review of the COMET database, and compares COS entries into the database including LMIC participants and HIC participants. While the topic area is of interest, I felt that the methods section needed to be presented more clearly and that some important points were missing from discussion. I have listed some points below, and would be happy to review another draft of this paper should the authors wish to revise and resubmit. • I wondered why the paper only describes COS published up to 2019? I would have suggested an updated search for new COS for inclusion just prior to paper submission, particularly given this is a relatively emerging area.• How were LMICs defined? For example, using classifications from the World Bank, or ?• The inclusion criteria and identification of relevant studies sections on page 5 were not clear. I would suggest a more clear presentation of your inclusion criteria, so that the search you undertook on the COMET database was directly reproducible. Include the fields searched, and your search timeframe here.• The text on lines 11-13 is repeated again on lines 17-19. Also, this is the first mention of PRO COS or core event sets – I thought this should have been included in the introduction where you introduce the concept of COS to the reader. The text on lines 24-26 re the annual update to the database is also repetitive from your introduction.• After reading your methods section I am not aware of how studies were screened from the database (for instance, was this done by a single reviewer or more than one etc).• Page 6, you state “There was a total of 552 references published by end of 2019 in the COMET Initiative database describing 370 COS”. Does this mean that the database allows for multiple references to a singular COS, or ? This needs to be better
--

explained in the introduction/methods around the COMET database – exactly what types of entries does a search of the database include?

- Are published COS completed COS, or do you just mean COS that are registered and published onto the database? And if so did you compare between COS that were completed (and potentially published as academic papers elsewhere) and those that had simply been registered as under preparation?
- You mention the PRISMA statement in your results, but this wasn't mentioned in the methods section. Did you follow PRISMA? If so, please include this detail into the methods.
- Page 5 – I felt that your statement “Given the differences in the global burden of diseases, which can potentially influence which COS are developed, and resources availability which can potentially determine the methods used in COS development, it is important to examine the differences/similarities across these two settings.” requires more explanation. This is crux of the issue in including participants from LMIC, and so should be clear in why this is important.
- Page 7, you state “...although it was assumed that in the absence of explicit mention the target population was adults, not children.” I am unclear as to whether this is an assumption that you the authors have made, or whether this is a database assumption? And if you have made this assumption, what did you base this on?
- I felt that your research finding of the % of COS as part of wider trial design vs specific for COS needed more introductory text to explain. For instance, you could introduce the difference when you first mention COS in your introduction. Perhaps a more clear introduction on the intended use for COS in the introductory text would be helpful too.
- I am not sure that I entirely follow your rationale re the finding that more COS are developed for NCDs in both HIC and LMIC. While disease burden in different contexts may be relevant to overall COS development, isn't this mostly about research funding priorities, particularly given that COS were predominantly intended for use in research in both settings? Perhaps COS would reflect disease burden in LMICs if they were more often being developed for clinical purposes, rather than research?
- The discussion on page 12, paragraph from line 4 to line 10 in regards to public participation would fit better into the results section. The discussion section around the barriers and facilitators to public participation should include issues around recruitment of CALD and other groups. There is plenty of literature in regards to this that could be referred to here. This seems to be the critical issue here, but is not explicitly discussed. How do researchers reach these population groups for inclusion in COS development studies etc.
- There is no mention of the link between the increase in LMIC participants and the publication of COS development guides, such as COS-STAD, the COMET handbook
- I don't quite follow the discussion re information asymmetry on page 12, line 21. Information asymmetry exists regardless of LMIC or HIC setting. Are you implying that information asymmetries are potentially greater in LMIC settings, and hence this is a bigger issue? This is not clear.

	 • What is the implication for studies that included only LMIC participants (of which there were only two)? I would have liked to have seen more discussion on this point. - Can you provide any discussion or comment on the level of uptake of COS between different settings (HIC and LMIC)? Some minor points:  • Page 1, line 24. The sentence starting with “Initially, the database...” is long and could be more succinctly phrased. Suggest splitting into two sentences. • Page 5, line 1. Delete include or had. Suggest delete had and change include to included. • Page 5, you state “There has been an almost two-fold increase in COS that have included participants from LMICs in the last five years [15% (35/229) of COS developed up to 2014 compared to 28% (40/141) 29 of COS developed between 2014 and 2019].” I would suggest a slight re-wording as this isn't quite correct considering you searched only up to the end of 2019 and it is now 2021 (i.e. the last five years would be 2016-2021). Suggest re-wording along the lines of: Between 2014 and 2019 there was a X% increase in COS that included participants from LMIC, as compared to the years preceeding 2014 (X%). • Page 6, line 22 – for consistency add % of COS only including HIC (80%). And % of COS with only LMIC participants. • Table 1 – in the far column add that brackets are %. Also round 10.3 as all other numbers are rounded. • Page 6 – can you add into lines 11-15 the % of studies in cancer and other disease domains.
--	---

REVIEWER	Fattah, Kazi University of Queensland Faculty of Humanities and Social Sciences, Sociology
REVIEW RETURNED	10-Apr-2021

GENERAL COMMENTS	This is a well written review article that addresses a very important matter. The authors have done an excellent job and I believe this article will be use for academics and practitioners working on developing/analysing core outcome sets. Have two very minor comments. One is that the conclusion is too brief and a bit of let down. Many readers often read the conclusion section immediately after reading the abstract and there isn't much in the conclusion here to encourage them to go through the whole article. The authors could briefly reiterate some of the key points from the findings and discussion in the conclusion which would be helpful for the reader. Second is that there are a few incorrect use of punctuations and types which needs to be taken care of. Can be easily done with a round of editing. Overall, it is a good read. Best wishes for the authors.
---

VERSION 1 – AUTHOR RESPONSE

Reviewer: 1

Dr. Victoria Brown, Deakin University Faculty of Health

- I wondered why the paper only describes COS published up to 2019? I would have suggested an updated search for new COS for inclusion just prior to paper submission, particularly given this is a relatively emerging area.

Response: The COS included in this review have been identified from the systematic review of COS, which is updated annually by the COMET Initiative. The review of 2020 COS studies is currently underway and the findings will be available later this year. In subsequent annual updates, the inclusion of LMIC stakeholders will be reported on as a follow-up to this review.

- How were LMICs defined? For example, using classifications from the World Bank, or ?

Response: LMICs were based on the classification of the Organization of Economic Co-operation and Development. It has now been clarified and inserted as a reference. See reference 22 (Organisation for Economic Co-operation and Development. Development Assistance Committee (DAC) list of Official Development Assistance (ODA) recipients [Accessed April 2020]. <http://www.oecd.org/dac/stats/daclist.htm>).

- The inclusion criteria and identification of relevant studies sections on page 5 were not clear. I would suggest a clearer presentation of your inclusion criteria, so that the search you undertook on the COMET database was directly reproducible. Include the fields searched, and your search timeframe here.

Response: The inclusion criteria and identification of relevant studies sections have been edited to make it clear that the studies included in the current review were identified from the original COMET systematic review and annual updates. Full details of the inclusion criteria and search strategy are provided in the COMET systematic review publications, but have been briefly summarised in the paper. The search strategy, which was developed for the original COMET systematic review, is provided in the appendix.

- The text on lines 11-13 is repeated again on lines 17-19. Also, this is the first mention of PRO COS or core event sets – I thought this should have been included in the introduction where you introduce the concept of COS to the reader. The text on lines 24-26 re the annual update to the database is also repetitive from your introduction.

Response: the description of PRO COS and core event sets has now been included in the background, “The types of studies included in the database are those in which COS have been developed, as well as studies relevant to COS development, including systematic reviews of outcomes and studies describing the development of a patient reported outcome (PRO) COS (a core set of patients reported symptoms and health-related quality of life domains) or core event set (a core set of adverse events or complications)”.
The repeated text has now been removed.

- After reading your methods section I am not aware of how studies were screened from the database (for instance, was this done by a single reviewer or more than one etc).

Response: We have now clarified (both above and in the Methods section) that studies were not identified from the COMET database, but rather from the annual COMET systematic reviews which are used to populate the database. The details of screening are provided in the papers that report these systematic reviews: The following references have been inserted in the manuscript.

Gargon E, Gurung B, Medley N, Altman DG, Blazeby JM, Clarke M, et al. Choosing important health outcomes for comparative effectiveness research: A systematic review. *PLoS One* 2014;9. <https://doi.org/10.1371/journal.pone.0099111>.

Gorst SL, Gargon E, Clarke M, Blazeby JM, Altman DG, Williamson PR. Choosing important health outcomes for comparative effectiveness research: An updated review and user survey. *PLoS One* 2016;11:1–12 [PubMed](https://pubmed.ncbi.nlm.nih.gov/) . <https://doi.org/10.1371/journal.pone.0146444>.

Gorst SL, Gargon E, Clarke M, Smith V, Williamson PR. Choosing Important Health Outcomes for Comparative Effectiveness Research: An Updated Review and Identification of Gaps. *PLoS One*. 2016;11(12):e0168403. <https://doi.org/10.1371/journal.pone.0168403>

Davis K, Gorst SL, Harman N, Smith V, Gargon E, Altman DG, et al. Choosing important health outcomes for comparative effectiveness research: An updated systematic review and involvement of low and middle income countries. *PLoS One* 2018;13:1–14 [PubMed](https://pubmed.ncbi.nlm.nih.gov/) . <https://doi.org/10.1371/journal.pone.0190695>.

Gargon E, Gorst SL, Harman NL, Smith V, Matvienko-Sikar K, Williamson PR (2018) Choosing important health outcomes for comparative effectiveness research: 4th annual update to a systematic review of core outcome sets for research. *PLoS ONE* 13(12): e0209869. <https://doi.org/10.1371/journal.pone.0209869>

Gargon E, Gorst SL, Williamson PR. Choosing important health outcomes for comparative effectiveness research: 5th annual update to a systematic review of core outcome sets for research. *PLoS One* 2019;14:e0225980–e0225980. <https://doi.org/10.1371/journal.pone.0225980>.

- Page 6, you state “There was a total of 552 references published by end of 2019 in the COMET Initiative database describing 370 COS”. Does this mean that the database allows for multiple references to a singular COS, or ? This needs to be better explained in the introduction/methods around the COMET database – exactly what types of entries does a search of the database include?

Response: This detail about the number of references contained within the COMET database has been removed from the Methods, as it is not relevant to the current study. Detail about the types of studies included in the COMET database has been added in the introduction section.

- Are published COS completed COS, or do you just mean COS that are registered and published onto the database? And if so did you compare between COS that were completed (and potentially published as academic papers elsewhere) and those that had simply been registered as under preparation?

Response: Published COS refer to any completed COS that are published in any academic papers. The COMET Initiative systematically search for all COS annually to ensure the database is kept up to date. We did not undertake a comparison of completed versus ongoing COS, as we do not have details of which COS are planning to include participants from LMICs.

- You mention the PRISMA statement in your results, but this wasn't mentioned in the methods section. Did you follow PRISMA? If so, please include this detail into the methods.

Response: The following information has been included in the methods section. “The review was conducted in accordance with the Preferred Reporting Items of Systematic Reviews and Meta-Analyses (PRISMA) guidelines and the included studies are shown in a PRISMA flow diagram (see figure 1)”

- Page 5 – I felt that your statement “Given the differences in the global burden of diseases, which can potentially influence which COS are developed, and resources availability which can potentially determine the methods used in COS development, it is important to examine the differences/similarities across these two settings.” requires more explanation. This is crux of the issue in including participants from LMIC, and so should be clear in why this is important.

Response: Text has been added to the explanation.

- Page 7, you state “...although it was assumed that in the absence of explicit mention the target population was adults, not children.” I am unclear as to whether this is an assumption that you the authors have made, or whether this is a database assumption? And if you have made this assumption, what did you base this on?

Response: In the table 1, we quantified the non-specified age. However, during the interpretation of the data, this was assumed to be adults. This was based on the disease being predominantly associated with the adult population and the lack of reference to any literature on children. The text has been edited to improve clarity.

- I felt that your research finding of the % of COS as part of wider trial design vs specific for COS needed more introductory text to explain. For instance, you could introduce the difference when you first mention COS in your introduction. Perhaps a clearer introduction on the intended use for COS in the introductory text would be helpful too.

Response: The following has been added as explanation in the introduction section. there are publications that describe a COS only (hereafter denoted specific for COS), and other publications that describe a COS as well as provides recommendations about other trial design aspects like eligibility criteria (hereafter denoted as COS as part of a wider trial design). This has also been added as footnotes to table 1.

- I am not sure that I entirely follow your rationale re the finding that more COS are developed for NCDs in both HIC and LMIC. While disease burden in different contexts may be relevant to overall COS development, isn't this mostly about research funding priorities, particularly given that COS were predominantly intended for use in research in both settings? Perhaps COS would reflect disease burden in LMICs if they were more often being developed for clinical purposes, rather than research?

Response: Thank you for this observation. We have added a plausible explanation that given that most COS are for research, and a majority of research funding is coming from HIC setting then the developed COS are likely to reflect research priorities from the HIC, of which NCDs are a major burden. Until we have more COS initiated from LMICs, we are still unlikely to see more COS matching the disease burden in LMICs.

- The discussion on page 12, paragraph from line 4 to line 10 in regard to public participation would fit better into the results section. The discussion section around the barriers and facilitators to public participation should include issues around recruitment of CALD and other groups. There is plenty of literature regarding this that could be referred to here. This seems to be the critical issue here, but is not explicitly discussed. How do researchers reach these population groups for inclusion in COS development studies etc.

- There is no mention of the link between the increase in LMIC participants and the publication of COS development guides, such as COS-STAD, the COMET handbook

Response: Thank you for this comment. We acknowledge that recruitment of CALD and other groups is an important part of ensuring that there is inclusivity in any research work. However, as an initial step our

objective here was to describe the inclusion of the public living in LMICs compared to those in HICs. This is distinct to looking at the inclusion of specific 'minority' groups in a given context or country, where much of the literature on the recruitment of CALD and other groups is focussed. Additionally, the description of public participation in the primary papers was sub-optimal and did not provide details on specific groups. Going forwards though we agree that overcoming barriers and facilitating the inclusion of minority groups in COS research is key.

The increase of the LMIC participation especially in the 'newer' COS could be partly attributed to the availability of guidance which might have enabled inclusion of wider stakeholder groups including LMIC participants. This has been added to the text.

- I don't quite follow the discussion re information asymmetry on page 12, line 21. Information asymmetry exists regardless of LMIC or HIC setting. Are you implying that information asymmetries are potentially greater in LMIC settings, and hence this is a bigger issue? This is not clear.

Response: Thank for your comment. We agree that information asymmetry is a common occurrence across HIC and LMICs. We do hypothesize that due to the differential literacy levels between HICs and LMICs, it is possible that the information asymmetry is probably a greater problem in LMICs than HICs. Additionally, presence of organized patient groups in HICs may enable patients to have more information on a given condition compared to those in LMICs where these groups are not very well defined/organized.

- What is the implication for studies that included only LMIC participants (of which there were only two)? I would have liked to have seen more discussion on this point.
- Can you provide any discussion or comment on the level of uptake of COS between different settings (HIC and LMIC)?

Response: Thank you for this comment. Unfortunately, with only two studies with exclusively LMIC participation, it would be inappropriate to provide any form of a conclusion on this point, however, this has informed the next piece of work where we are looking at the barriers and facilitators for inclusion of LMIC stakeholders in COS development including initiation of COS development from LMIC settings. Additionally, in the wider work related to this review, we are looking to describe strategies that will potentially increase the uptake of COS in LMIC which is still suboptimal.

Some minor points:

- Page 1, line 24. The sentence starting with "Initially, the database..." is long and could be more succinctly phrased. Suggest splitting into two sentences.

Response: This has been edited.

- Page 5, line 1. Delete include or had. Suggest delete had and change include to included.

Response: Text amended.

- Page 5, you state "There has been an almost two-fold increase in COS that have included participants from LMICs in the last five years [15% (3/229) of COS developed up to 2014 compared to 28% (40/141) 29 of COS developed between 2014 and 2019]." I would suggest a slight re-wording as this isn't quite correct considering you searched only up to the end of 2019 and it is now 2021 (i.e. the last five years would be 2016-2021). Suggest re-wording along the lines of: Between 2014 and 2019 there was a X% increase in COS that included participants from LMIC, as compared to the years preceeding 2014 (X%).

Response: Text amended.

- Page 6, line 22 – for consistency add % of COS only including HIC (80%). And % of COS with only LMIC participants.

Response: Amended

- Table 1 – in the far column add that brackets are %. Also round 10.3 as all other numbers are rounded.

Response: Corrected

- Page 6 – can you add into lines 11-15 the % of studies in cancer and other disease domains.

Response: Text amended.

Reviewer: 2

Mr. Kazi Fattah, University of Queensland Faculty of Humanities and Social Sciences

Response: Thank you for your feedback. We have edited the text in response.

VERSION 2 – REVIEW

REVIEWER	Brown, Victoria Deakin University Faculty of Health, Deakin Health Economics
REVIEW RETURNED	17-Jun-2021
GENERAL COMMENTS	Thank you to the authors for addressing the reviewer comments. The paper is significantly strengthened, and the methods and purpose have become much more clear to the reader. Delete "to" page 5, line 16.